# Programmable site-selective labeling of oligonucleotides based on carbene catalysis

Yang-Ha Lee[1], Eunsoo Yu [1] & Cheol-Min Park [1✉]

Site-selective modification of oligonucleotides serves as an indispensable tool in many fields of research including research of fundamental biological processes, biotechnology, and nanotechnology. Here we report chemo- and regioselective modification of oligonucleotides based on rhodium(I)-carbene catalysis in a programmable fashion. Extensive screening identifies a rhodium(I)-catalyst that displays robust chemoselectivity toward base-unpaired guanosines in single and double-strand oligonucleotides with structurally complex secondary structures. Moreover, high regioselectivity among multiple guanosines in a substrate is achieved by introducing guanosine-bulge loops in a duplex. This approach allows the introduction of multiple unique functional handles in an iterative fashion, the utility of which is exemplified in DNA-protein cross-linking in cell lysates.

[1] Department of Chemistry, UNIST (Ulsan National Institute of Science & Technology), Ulsan, Korea. ✉email: cmpark@unist.ac.kr

Sequence-specific DNA-binding proteins such as transcription factors (TFs), polymerases, and DNA modification enzymes play a crucial role in regulating gene expression and maintaining various cellular functions. Therefore, the identification of these proteins and elucidation of their functions have drawn a great deal of attention[1–4]. The transient interactions between DNA and the corresponding binding proteins led to the development of various covalent trapping methods including diazirine-derived carbenes[5] and UV-based cross-linking[6,7]. Also, successful trapping of DNA-binding proteins has been demonstrated by employing self-ligating ligands (HaloTag, SNAP, CLIP) fused to the proteins of interest and DNA probes linked to the corresponding tags[8]. These indispensable methods for the study of DNA-binding proteins require chemical modification of natural nucleic acids to introduce functional handles, often with more than one functionalities for sophisticated manipulations including cross-linking and detection.

The current standard method for the site-selective introduction of orthogonal functionalities is the prior synthesis of pre-functionalized nucleoside phosphoramidite monomers via laborious multistep synthesis, followed by solid-phase oligonucleotide (ON) synthesis[9–12]. More importantly, the ability to introduce multiple non-redundant functional groups is limited by the availability of chemically compatible orthogonal protecting groups that can be removed selectively while preserving the labile glycosidic bonds in oligonucleotides intact.

Alternatively, post-synthetic modification of ONs may provide more efficient access to site-selective functionalization. While elegant methods based on enzyme-mediated and chemical-based modifications have been reported to date[13–15], significant challenges still remain. For example, those tailored for the modification of terminal sites include oxidative cleavage of ribose 3′-end[16,17], 3′-phosphorylation by deoxyribozyme-mediated phosphate transfer[18], 5′-phosphoramidation of 5′-phosphate via phosphorimidazolide[19], and 5′-phosphate alkylation by diazo compounds[20].

Meanwhile, internal sites-specific alkylation of bases has been developed. The SELEX (systematic evolution of ligands by exponential enrichment)-based modifications include the ribozyme-catalyzed protocol[21] and self-alkylation of RNAs by employing high affinity electrophilic ligands[22–24]. Also, proximity-driven modifications by hybridizing opposite strands bearing reactive groups have been reported[25–27]. Enzymes such as methyltransferase and tRNA-guanine transglycosylase have been employed for selective modification[28,29], which necessitates specific sequences for enzyme recognition and laborious engineering of enzymes/cofactors. Also, 2′-OH modification of RNAs has been shown by using acyl imidazole reagents[30,31] and SELEX-based ribozymes[32].

The unique reactivity of metal carbene complexes has led to the development of diverse synthetic applications including bioconjugation[33–35]. Their use for the modification of proteins has been reported by several research groups[36–39]. Recently, in their pioneering work, Gillingham and co-workers reported efficient post-synthetic modification of ONs based on Rh(II)-carbenes, which was, however, lacking base selectivity. Later, the development of Cu(I)-carbenes allowed the chemoselectivity for guanosine (G) bases, in which the limited substrate scope of sterically unhindered single strands remained to be addressed[40–42]. While the chelation-driven selectivity of Cu(I)-catalysts works smoothly on simple substrates, complex substrates/matrices bearing various multiple chelation sites pose significant challenges to this type of catalysts regarding catalyst turnover (See Supplementary Fig. 1 for the comparison of the previous and this works).

In our efforts to develop a catalytic system that allows site-selective modification among multiple Gs embedded in structurally complex oligonucleotides, we aim to address the two fundamental selectivities toward substrates; chemo- and regio-selectivity. In this regard, we reason that the discovery of a chelation-free catalyst is crucial to overcome the issue of catalyst turnover in such a highly chelating environment, in which the chemoselectivity among the different nucleotide bases needs to be driven by the intrinsic reactivity of a catalyst, instead of a chelation effect. Secondly, we envisage that the regioselectivity among multiple Gs could be realized by the introduction of G-bulge loops in duplexes, in which the difference in reactivity between base-paired and unpaired Gs in a duplex may provide the regioselectivity. The prerequisite for this approach is the identification of a catalyst, which displays robust catalytic activity toward G-bulges located in structurally complex duplexes. The successful development of this approach would allow a programmable synthesis of ONs with multiple unique functional groups in an iterative fashion by simple replacement of the corresponding opposite strands (Fig. 1a).

Here, we describe the post-synthetic modification of oligonucleotides at unpaired Gs in a chemo- and regio-selective manner based on rhodium(I)-carbene catalysis. The ability of the protocol for the selective modification of G-bulges allows the introduction of multiple unique functional handles in an iterative fashion, the utility of which is exemplified in DNA-protein cross-linking in cell lysates.

## Results and discussion

**Optimization and substrate scope.** To identify an optimal catalyst for the selective modification of nucleobases, we commenced with screening of various metal catalysts (5 mol%)[43–45] in the presence of diazoacetone and a mixture of four nucleosides as a model substrate for base-selectivity in 50% THF-H$_2$O (Fig. 2, Entries 1 and 2; Supplementary Table 2, Entries 5–9), however, all of which turned out to be ineffective. Gratifyingly, rapid conversion was observed with exclusive formation of a product within 30 min when [Rh(COD)Cl]$_2$ was employed as a catalyst (45%, Fig. 2, Entry 3). The structure was unambiguously assigned as O$^6$-acetonyl deoxyguanosine (dG) **6** based on heteronuclear multiple bond correlation (HMBC) spectroscopic analysis (correlation between C6–H9, Fig. 3a). Since the secondary structures of ONs may be affected by the presence of an organic solvent, we examined the reaction in a reduced fraction of THF (10% THF-H$_2$O), and found that it afforded a comparable yield compared to that in 50% THF-H$_2$O (96% vs. 93%, respectively, Supplementary Table 2, Entry 1 and 2). We also confirmed by performing a reaction with **7b** in pure aqueous buffer that 10% THF does not affect the reaction (Supplementary Fig. 5). The effect of pH revealed a higher conversion at pH = 6.0 than pH = 7.4 (Supplementary Table 2, Entry 3 and 4). Eventually, the reaction conditions were optimized as 10 mol% catalyst loading in 10% THF-H$_2$O at pH = 6.0, MES buffer (98%; Fig. 2, Entry 4).

Encouraged by the excellent catalytic activity of [Rh(COD)Cl]$_2$, an extensive screening with various Rh(I) salts was performed, however, other Rh(I) catalysts turned out to be less effective (Fig. 2, Entries 5–8). Likewise, an examination of various diazo compounds revealed that diazoacetone possesses unique reactivity that affords excellent yield (Fig. 2, Entries 4 vs. 9; see Supplementary Table 2 for a complete list of diazo compounds and buffer conditions). Deoxyinosine (dI), a structural analog of dG, was also examined for its reactivity relative to dG and was found to be slightly less reactive (dG/dI = 1.3: 1, Supplementary Fig. 2).

The optimized conditions feature an optimal balance between the reactivity/selectivity toward Gs and stability in aqueous buffer media. Also, the advantages of diazoacetone as a carbene precursor include the keto group serving as an orthogonal

## a

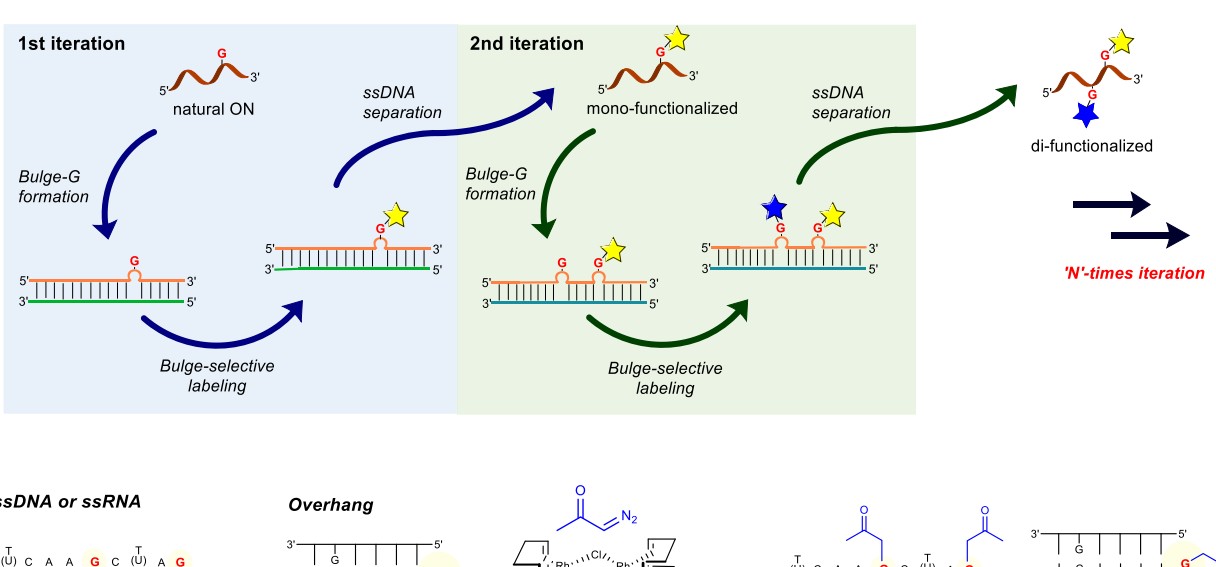

## b

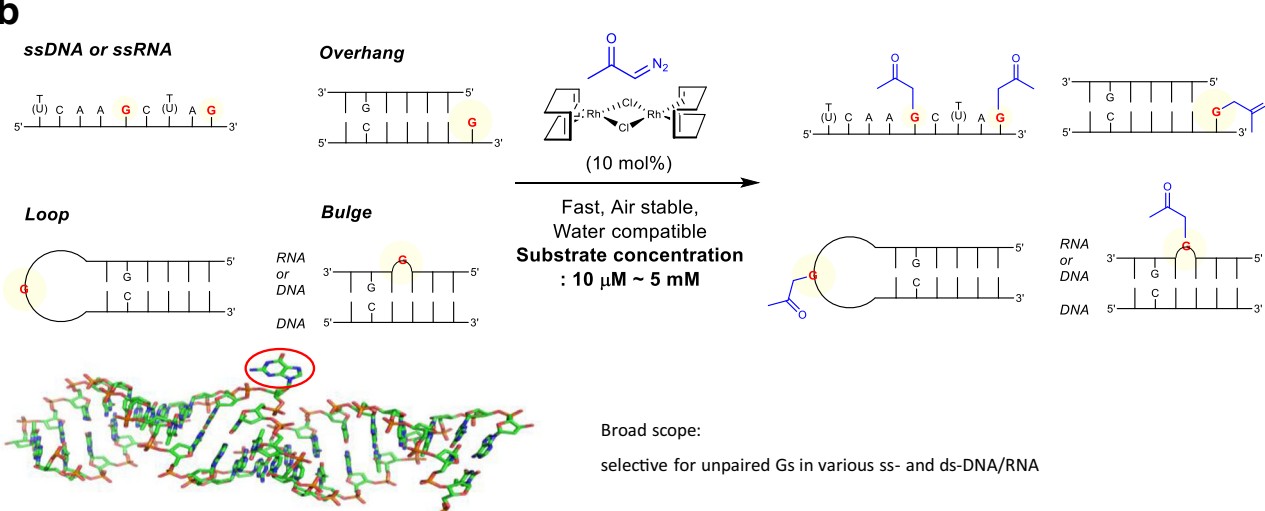

**Fig. 1 Site-selective modification of oligonucleotides. a** Iterative functionalization of oligonucleotides. **b** Compatibility with various secondary structures in a wide range of concentrations; 3D model of bulge-guanosine (G) dsDNA.

| Entry | Catalyst | Time | Yield[a] | Entry | Catalyst | Time | Yield[a] |
|---|---|---|---|---|---|---|---|
| 1[b] | Rh₂(OAc)₄ | 24 h | N/R | 6[c] | [Rh(COD)(MeCN)₂]BF₄ | o/n | 76% |
| 2[b] | Rh₂(TFA)₄ | 24 h | N/R | 7[c] | [Rh(COD)₂]OTf | o/n | 48% |
| 3[b] | [Rh(COD)Cl]₂ | 30 min | 45% | 8[c] | Rh(CO)₂(acac) | o/n | N/R |
| 4[c] | [Rh(COD)Cl]₂ | 10 min | 98% | 9[d] | [Rh(COD)Cl]₂ | 3 h | N/R |
| 5[c] | Rh(COD)(acac) | 60 min | 83% | | | | |

**Fig. 2 Optimization of Rh(I)-catalyzed O⁶-G labeling.** [a]Based on HPLC analysis. [b]Conditions a: each nucleoside (5 mM), catalyst (5 mol%), diazoacetone (1.2 equiv.), 50% THF-H₂O, r.t. [c]Conditions b: each nucleoside (5 mM), catalyst (10 mol%), diazoacetone (8 equiv.), 10% THF-H₂O, MES (20 mM), pH 6.0, r.t. [d]Used ethyl diazoacetate instead of diazoacetone (deviation from Conditions a). N/R = no reaction. o/n = overnight.

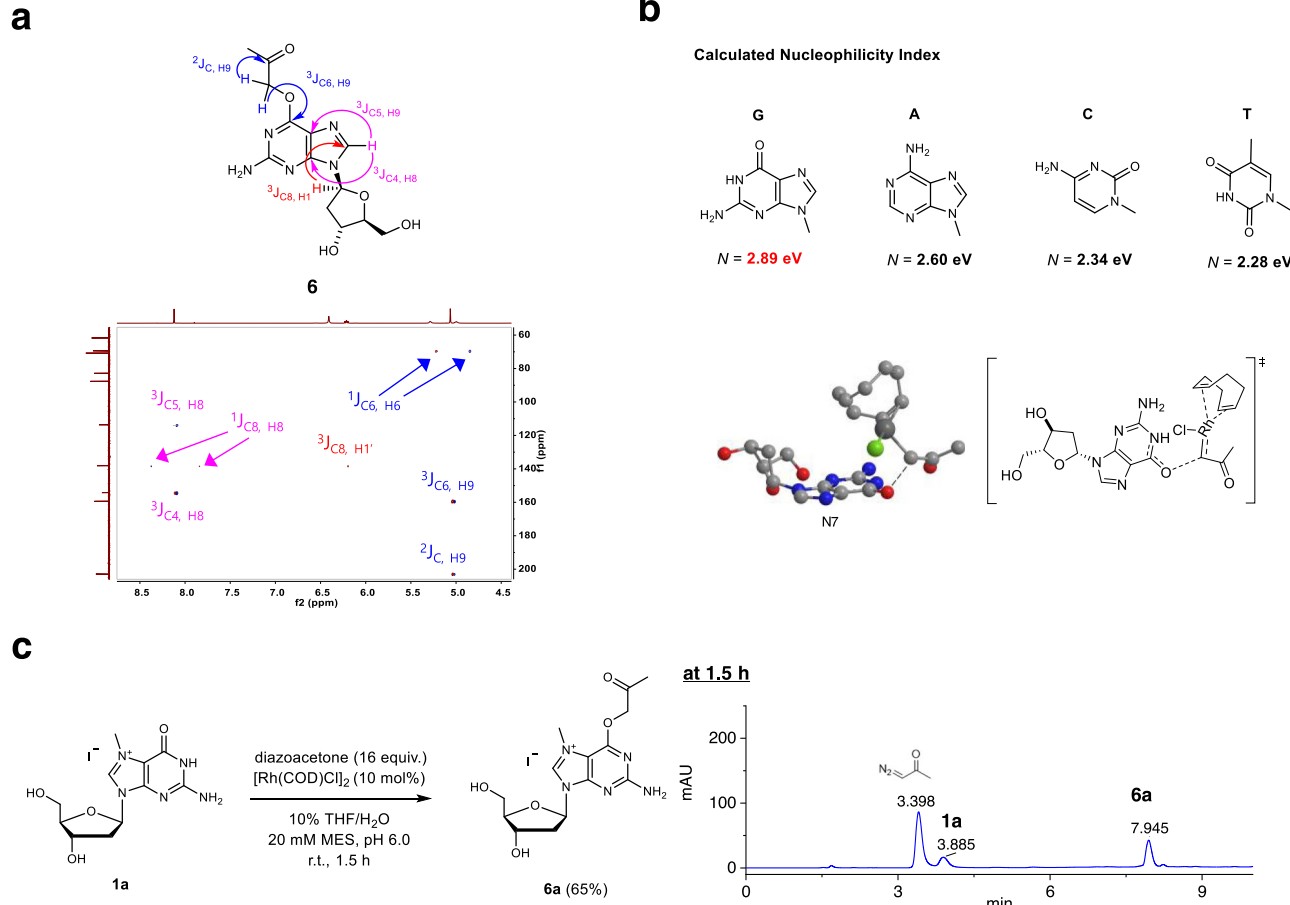

**Fig. 3 Structure and mechanistic analysis of Rh(I)-catalyzed O⁶-G labeling. a** Expanded $^1$H-$^{13}$C HMBC (DMSO-$d_6$) spectrum of **6**. **b** Calculated nucleophilicity index of four canonical nucleobases (A, T, G, and C); transition state geometry of the catalyst approaching guanosine obtained by DFT calculations. **c** Reaction performed with N$^7$-methyl-2′-deoxyguanosine and its HPLC trace.

functional handle, minimal steric disruption owing to its small size crucial for the studies involving protein-nucleic acid binding interactions, and the high aqueous solubility of diazoacetone. In addition, the by-product hydroxyacetone, which forms inevitably owing to the reaction of carbenes with water, could be readily removed by simple evaporation to facilitate the purification step.

To gain insight on the selectivity toward guanosine among the nucleosides, we performed density function theory (DFT) calculations. Based on the nucleophilicity $N$ index, which was recently developed to describe the behavior of various nucleophiles[46,47], we calculated them for the four bases (Fig. 3b). It turned out that guanine showed the highest value on the nucleophilic scale, which is in good correlation with the observed selectivity. Also, the complete energy profile of the reaction pathway was calculated (Supplementary Fig. 62). Overall, the reaction is highly exergonic with −30.7 kcal/mol. The highest activation barrier turned out to be the initial rhodium carbene formation with 11.6 kcal/mol followed by a barrier of 6.4 kcal/mol for the O6-C9 bond formation. The transition state geometry displays the chelation-free approach of the carbene complex toward the guanosine O⁶ supporting the unattenuated reactivity in the presence of numerous chelating functionalities. To further support the chelation-free model, we performed a reaction with N⁷-methyl-2′-deoxyguanosine, which is unable to make chelation, and found that it provided the corresponding O⁶-acetonyl product **6a** in 65% yield (Fig. 3c; Supplementary Fig. 22 for MS analysis; SI p.104 for NMR analysis after sugar was hydrolytically removed).

With the optimized conditions at the monomer level, we examined the selectivity of the carbene complex toward Gs embedded in ONs. Mg(OAc)$_2$ was included for subsequent applications to duplexes in mind, which stabilizes them by shielding the phosphate backbone charges[48]. The G-selectivity was confirmed by tandem mass spectrometry by employing tetramer **7a** as a model system (Fig. 4a, b).

Since the reaction site, guanosine O⁶, is engaged in the base pairing, we reasoned that it might provide us another level of selectivity between paired and unpaired Gs. It turned out that exclusive modification at the unpaired G was achieved while those with base pairing remaining intact when hairpin **7b** containing multiple Gs in the stem and overhang was subjected to the reaction. The site of modification was confirmed by digestion with endonuclease EcoRI followed by identification of the acetonylated fragment by LC–MS analysis (see Fig. 4c, d). In addition, site-selectivity of the modification was further confirmed by a primer extension assay with **8h** (Supplementary Fig. 6).

Based on these results, we investigated the substrate scope with hairpins, duplexes, and hybrid duplexes (Fig. 5 and Supplementary Table 4). Beforehand, a brief examination of single-stranded ONs confirmed the selective modification of Gs with good efficiency (Supplementary Table 3). Next, we expanded our investigation on various duplex ONs (Fig. 5). The influence of base pairing on the reactivity of the catalytic system in several hairpin and duplex ONs bearing Gs at various positions including overhang, loop, base-paired, and mismatched was examined.

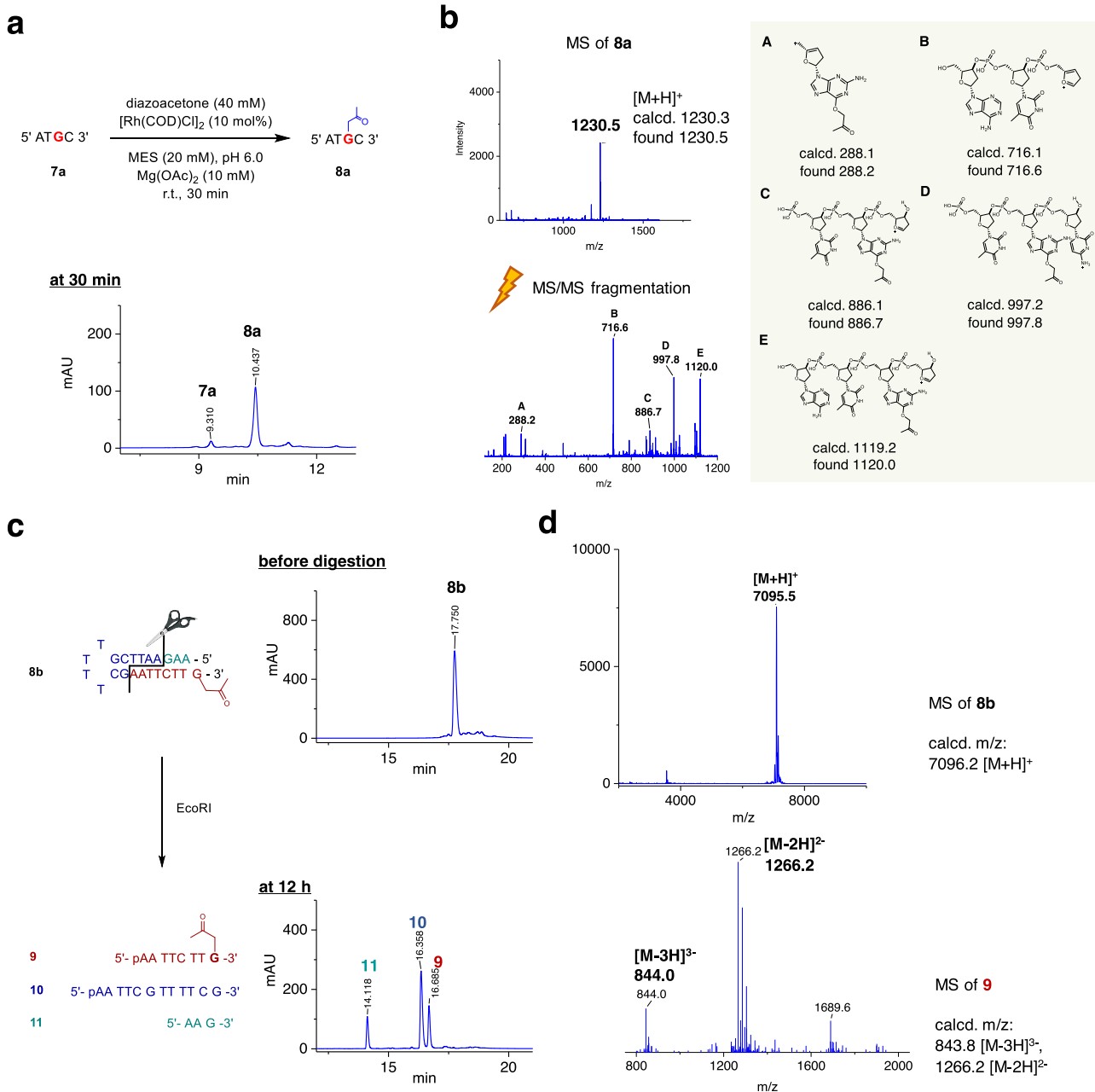

**Fig. 4 Structure analysis of Rh(I)-catalyzed O⁶-G labeling of oligonucleotides. a** HPLC analysis of the acetonylation of **7a**. **b** Identification of G modification by MALDI-TOF/TOF analysis of **8a**. Mass spectrum (MS) of **8a** and its MS/MS fragmentation with labeled fragments. **c** Endonuclease digestion of acetonylated hairpin oligonucleotide **8b** and the corresponding HPLC spectra for the analysis of the modification site. **d** MS of **8b** and **9**.

Overall, the reaction turned out to be highly selective toward unpaired Gs regardless of hairpins or duplexes **7b**–**7j** (Fig. 5). Hairpin **7e** bearing two Gs at both 5′- and 3′-ends reacted to give the corresponding dual labeling product in 60% yield. Efficient conversions were observed with an overhang G located at either 3′- or 5′-end of dsODN **7f** and **7g**. In addition, dsODN (double-stranded oligodeoxynucleotide) **7h** containing a G at an internal position of a long overhang participated well to give an excellent yield (89%) showing highly general reactivity. Moreover, RNA hairpin **7i** and **7j**, which have unpaired G at the overhang and loop positions, gave excellent yields (88 and 83%, respectively).

Although the metal-carbene complex displays a high base-selectivity toward Gs, its lack of site-selectivity on ONs bearing multiple unpaired G residues poses severe limitation. To address the problem, we initially envisaged that the introduction of

mismatched Gs would provide selectivity over those engaged in base-pairing. It turned out that the reaction on hairpin **7k** containing a mismatch at an internal position was sluggish (26%, Fig. 5). Based on the speculation that the stack-in conformation of mismatched Gs may interfere the catalyst accessibility, we reasoned that the introduction of Gs as a bulge loop may provide a better exposure toward the catalyst. We were gratified to find that the reaction proceeded with an exclusive modification at the G-bulge in excellent yield in 30 min when tested on the 20-mer and 25-mer bulge duplexes **7l** and **7m**. Dual labeling of duplex **7n**, in which both strands contain G-bulges, proceeded efficiently. Also, a comparable yield was obtained with hairpin **7o**. A control experiment with dsODN **7p** lacking a G-bulge showed no reaction, which further confirms the selectivity. We also investigated oligoribonucleotides (ORNs) as substrates. G-bulge

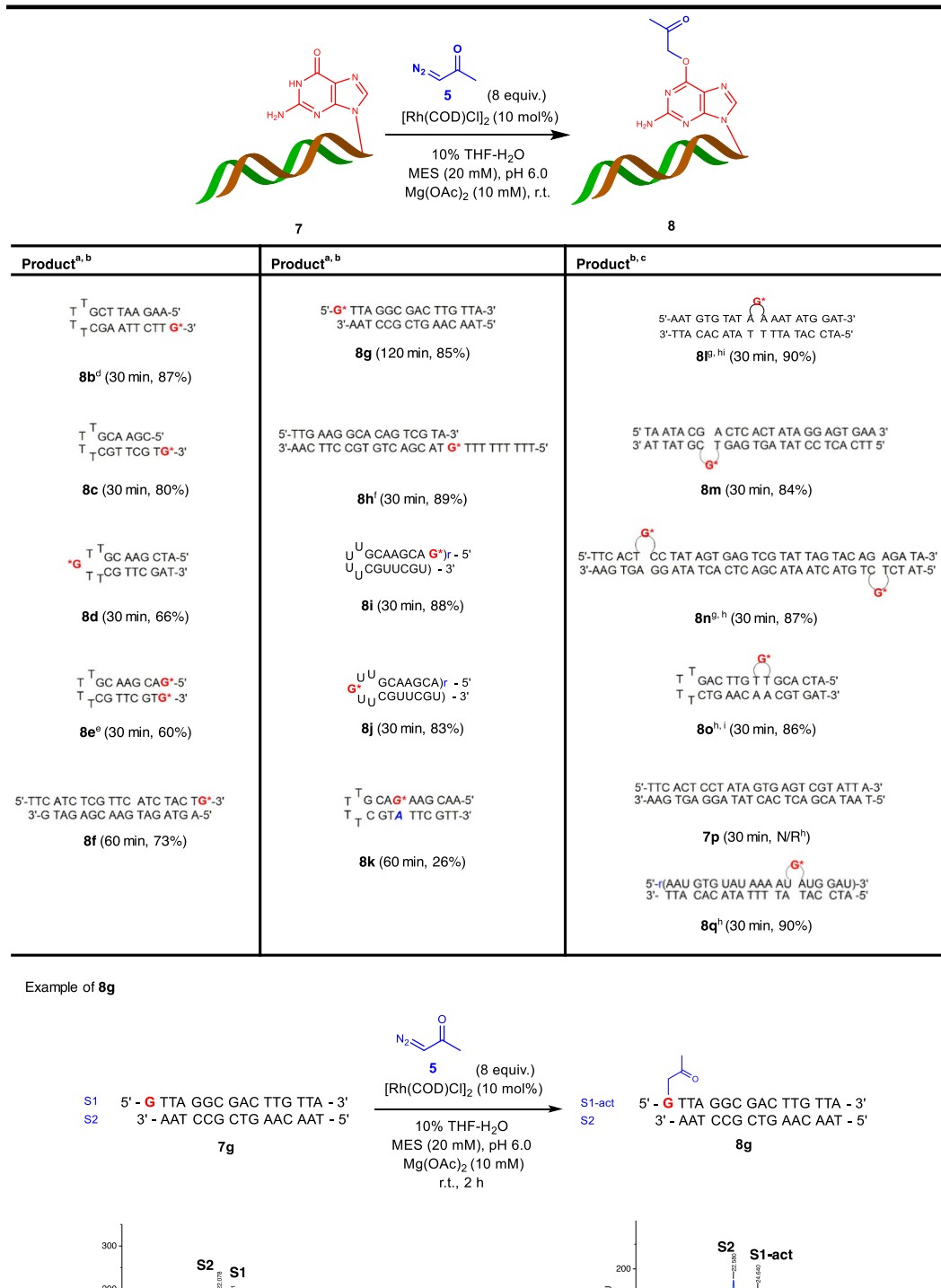

**Fig. 5 Scope for hairpins and duplexes.** All oligonucleotide (ON)s are oligodeoxynucleotides (ODN)s unless noted as r(sequence). All the reactions were analyzed by HPLC, as exemplified at the bottom of the figure. In **8k**, mismatched A was marked in blue. [a]Conditions a: ON (5 mM), diazoacetone (8 equiv.), [Rh(COD)Cl]₂ (10 mol%), 10% THF-H₂O, MES (20 mM), pH 6.0, Mg(OAc)₂ (10 mM), r.t. [b]Yield was calculated by HPLC analysis. [c]Conditions b: ON (5 mM), diazoacetone (100 mM), [Rh(COD)Cl]₂ (10 mol%), 10% THF-H₂O, MES (50 mM), pH 6.0, Mg(OAc)₂ (50 mM), r.t. [d]Modification site was determined by endonuclease digestion (See Fig. 4c, d). [e]16 equiv. of diazoacetone, 15 mol% [Rh(COD)Cl]₂ (deviation from Conditions a). [f]16 equiv. of diazoacetone (deviation from Conditions b). [g]40 equiv. of diazoacetone (deviation from Conditions b). [h]Yield was calculated after oxime ether conjugation with excess amount of benzyloxyamine. [i]12 equiv. of diazoacetone (deviation from Conditions b). N/R = no reaction.

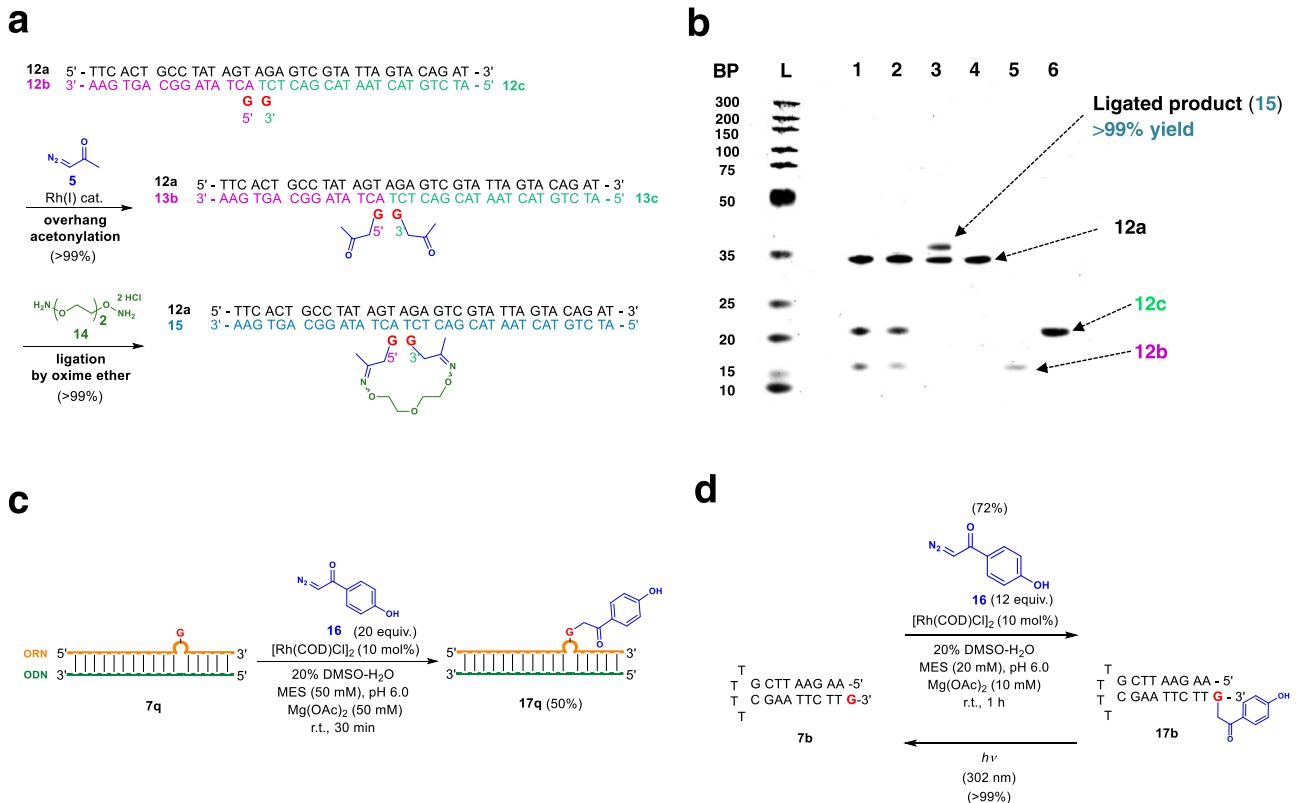

**Fig. 6 Chemical ligation of oligodeoxynucleotides and single-step synthesis of photocaged oligonucleotides. a** Synthetic scheme for the ligation of oligonucleotides with an oxime ether linker. **b** Denaturing PAGE analysis of the ligation visualized by SYBR gold. Lanes; L: DNA ladder, 1: before acetonylation, 2: after acetonylation, 3: after ligation, 4: **12a**, 5: **12b**, 6: **12c**. **c** Synthesis of photocaged single-stranded oligoribonucleotide based on the G-bulge protocol. **d** Caging and decaging of **17b**. Source data are provided as a Source Data file.

ORN **7q** was prepared by ORN-ODN hybrid duplex formation and was subjected to the standard conditions. Gratifyingly, the reaction proceeded to give **8q** in 90% yield.

**Chemical ligation of ODNs and single-step synthesis of photocaged ONs.** Chemical ligation of ODNs with unnatural linkers allows the preparation of the corresponding elongated ODNs, on which diverse functional groups could be attached. To this end, we prepared a duplex comprised of the template sequence **12a** paired with two ODN fragments **12b** and **12c** bearing a G-overhang on each fragment (Fig. 6a). The reaction with diazoacetone selectively proceeded at the two G-overhang residues to give **13b** and **13c** in quantitative yields, each of which was functionalized with an acetonylated-G (act-G) (see SI p. 33 and Supplementary Fig. 53 for details). Ligation was performed on the act-G duplex ODN with dialkoxyamine linker **14**. The ligation proceeded in quantitative yield confirmed by LC–MS and denaturing PAGE analyses (Fig. 6b; see Supplementary Fig. 54); the low mobility band slightly above the template sequence **12a** corresponds to the ligated product **15**.

Photocleavable protecting groups serve as a powerful tool in many biological studies[49–52]. The typical synthesis of photocaged ONs begins with the preparation of phosphoramidite monomers functionalized with a photocleavable group, which usually involves lengthy syntheses. For example, Woodson and coworkers[9] reported the temporally controlled Hfq-catalyzed annealing study by using photocaged ORNs. Thus, to examine the applicability of our method, G-bulge ORN **7q** prepared by ORN-ODN hybrid duplex formation was reacted with α-diazo-*p*-hydroxyacetophenone **16** (20 equiv.) in the presence of 10 mol% [Rh(COD)Cl]₂ (20 % DMSO-H₂O, MES, pH = 6.0). We found

that the reaction smoothly proceeded to afford **17q** in 50% yield (Fig. 6c, Supplementary Fig. 55). Moreover, decaging of **17b** was performed to regenerate **7b** in quantitative yield (Fig. 6d, Supplementary Fig. 59). These results demonstrate the efficiency of the single-step synthesis of photocaged ORNs and ODNs.

**DNA-protein cross-linking (DPC) with a DNA sequence-specific binding protein.** A programmable method enabling site-selective introduction of multiple unique functional handles in an iterative fashion on various ONs is highly desirable for the identification of numerous nucleic acid-binding proteins. As such, we sought to apply our protocol for the covalent trapping of sequence-specific DNA-binding proteins with the corresponding modified dsODNs (Fig. 7a). To examine the feasibility, we chose T7 RNA polymerase (RNAP), which possesses specificity for the T7 promoter sequence and took a two-step approach; (a) we first confirmed the cross-linking in the absence of cell lysate with a dsON bearing single acetonyl group for covalent trapping via reductive amination with an adjacent lysine residue in the binding site of RNAP. (b) Subsequently, we performed the cross-linking in the presence of cell lysate with a doubly modified dsON, in which one acetonyl group was used for covalent trapping and the other for the introduction of a fluorescent probe.

To assess the impact of the location of modification, we designed two dsDNA probes bearing a G-bulge on the antisense strands located either in the middle of or several nucleotides away from the T7 promoter sequence (Fig. 7b). The act-G-bulge dsDNAs were prepared by our standard protocol. Reductive amination was performed by incubation of T7 RNAP with the act-G-bulge dsDNAs in the presence of NaCNBH₃, and the samples were analyzed on SDS–PAGE gel visualized by silver and

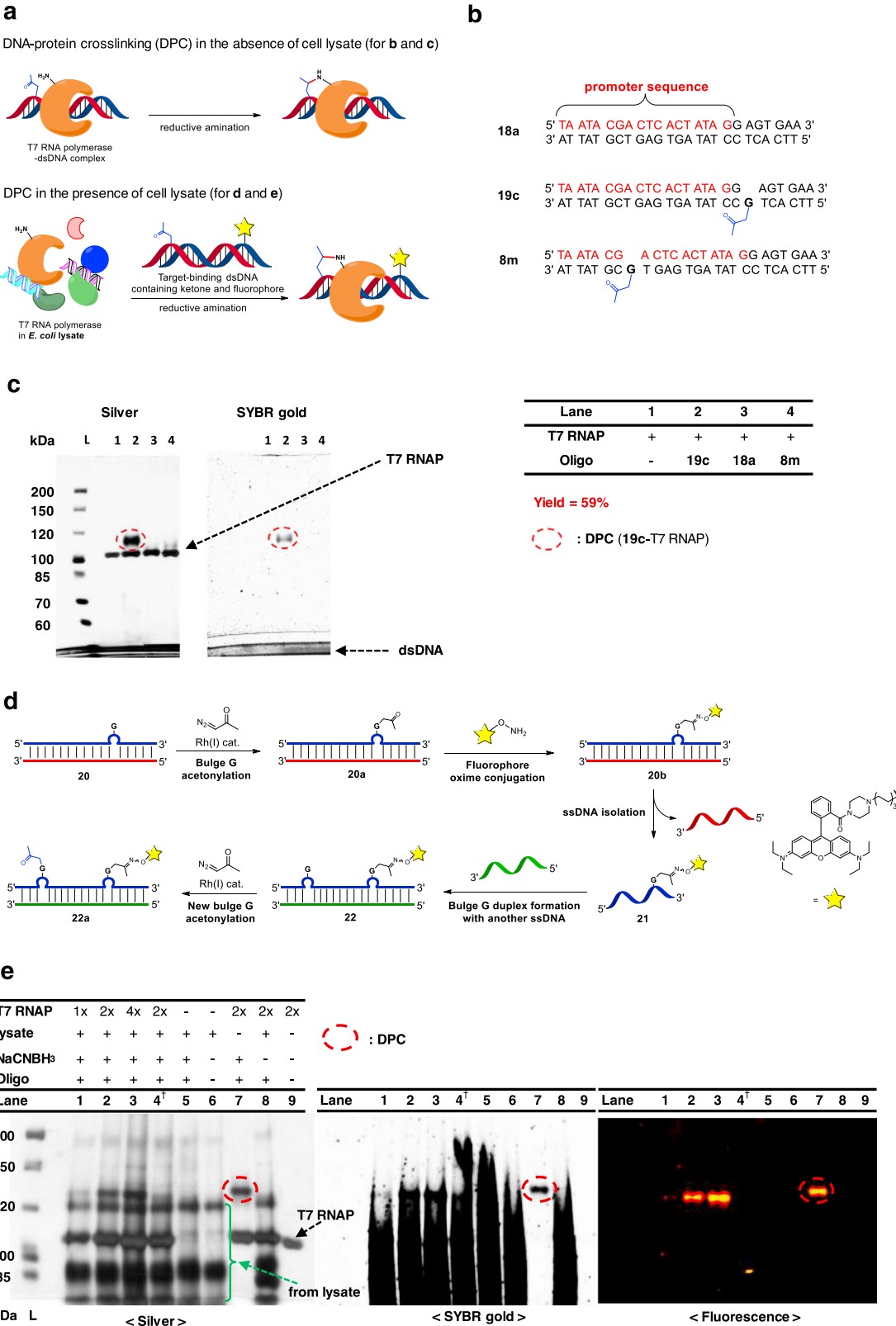

**Fig. 7 Preparation of dual labeled dsDNA and T7 RNA polymerase-promoter cross-linking. a** DNA-protein cross-linking of the probe and DNA-binding protein in the absence (**b** and **c**) and in the presence of *E. coli* lysate (**d** and **e**). **b** Oligonucleotide probes employed for **c**. **c** SDS–PAGE analysis of the cross-linking reaction visualized by silver and SYBR gold stains. Conditions: dsDNA (20 μM), T7 RNA polymerase (T7 RNAP) (6.8 μM), HEPES (40 mM), pH 7.4, Mg(OAc)$_2$ (5 mM), NaCNBH$_3$ (100 mM), 37 °C, 12 h. **d** Synthetic scheme for dual G-bulge probe **22a** employed for **e**. **e** SDS–PAGE analysis of DNA-protein cross-linking in *E. coli* lysate. Conditions: dsDNA (20 μM), T7 RNAP (3.4 μM, for 1x), HEPES (40 mM), pH 7.4, Mg(OAc)$_2$ (5 mM), NaCNBH$_3$ (100 mM), 30% *E. coli* lysate, 37 °C, 12 h. †unmodified oligo was used. L: protein standard ladder. Source data are provided as a Source Data file.

SYBR gold stains (Fig. 7c). Whereas strong bands corresponding to both the cross-linked product and T7 RNAP were observed for **19c** in the silver staining, **8m** bearing a bulge in the middle of the promoter sequence showed only the band for RNAP with no cross-linking (lanes 2 and 4, respectively). We speculate that the failure of **8m** in cross-linking may have resulted from the disruption of the binding of RNAP caused by the bulge or the absence of a proximal lysine residue. The presence of dsODN **19c** in the cross-linked product was further confirmed by SYBR gold staining (lane 2), in which the band matched with the upper band in the silver stain (lane 2), as opposed to the absence of the corresponding band for **8m** (lane 4, SYBR gold). Moreover, no cross-linking was observed in the reductive amination when a natural duplex **18a** (lane 3) or a duplex **19i** bearing a non-promoter sequence (Supplementary Fig. 10) was employed as negative controls. These results establish that the cross-linking is sequence-specific and modification site-specific.

Next, we examined the cross-linking by employing *E. coli* whole cell lysate as a complex matrix (Fig. 7a). To detect the cross-linked product in the cell lysate, we designed a dual G-bulge dsDNA probe, in which the second acetonyl group was used for conjugation with a fluorescence probe (Fig. 7d; see SI p. 35, Supplementary Figs. 60 and 61). For the selective manipulation of the two acetonyl groups, they were sequentially introduced in an iterative manner. Thus, the formation of a first G-bulge dsDNA at the desired position was achieved by using the corresponding opposite strand, which was subsequently reacted with diazoacetone to give **20a**. After Rhodamine B alkoxyamine was linked to the ketone via an oxime ether, the ssDNA **21** bearing the fluorophore was isolated. Introduction of a second G-bulge by duplex formation with the corresponding opposite strand followed by the reaction with diazoacetone afforded the dual act-G-bulge dsDNA probe **22a** ready for the cross-linking experiment. Notably, the reaction proceeded selectively at the bulge-G while the dye moiety remaining intact. The stability of the duplex **22** may be slightly lower than **20** owing to an additional mismatch pair, however, it does not appear to affect the reaction based on the observation that the second labeling also proceeds uneventfully.

Cross-linking was performed by reductive amination, in which **22a** was incubated with NaCNBH$_3$ in the presence of *E. coli* lysate pretreated with T7 RNAP (Fig. 7e, lanes 1–3). Control experiments were also performed (a) with nonmodified form of **22a** (lane 4), (b) without T7 RNAP (lane 5), (c) with lysate only (lane 6), (d) without lysate (lane 7), (e) without NaCNBH$_3$ (lane 8), (f) with T7 RNAP only (lane 9). The SDS–PAGE gel was visualized by fluorescence, silver stain, and SYBR gold. The visualization with fluorescence confirmed that selective cross-linking was successfully achieved even in the presence of the cell lysate (lanes 1–3); the bands with the same mobility compared to that in the positive control experiment (lane 7) was detected.

To confirm that the band corresponding to the cross-linking of **22a** with RNAP is indeed derived from the sequence-specific conjugation among the proteins in the lysate, we performed a DPC experiment in the presence of varying concentrations of RNAP (lanes 1–3). The observation that the intensity of the fluorescent bands on the SDS–PAGE gel increases with the increasing concentration of RNAP clearly supports that the bands are derived from the sequence-specific conjugation of RNAP with **22a**. Further staining with silver and SYBR gold also provided consistent results with the expected stain patterns (lanes 1–3 vs. positive control 7). Moreover, the results from several negative control experiments further support the specific cross-linking (lanes 4–6, 8, and 9).

In summary, we developed chemo- and regio-selective modification of ONs based on Rh(I)-carbene catalysis, which

allows the selective introduction of an acetonyl group at G-bulges located in various ssODNs, ssORNs, hairpins, duplexes, and hybrid duplexes. Moreover, the protocol enables the introduction of multiple non-redundant functional groups in an iterative fashion. The utility of the protocol has been demonstrated with several applications.

## Methods

**Rh(I)-catalyzed O$^6$-G acetonylation of ODNs or ORNs.** All reactions were performed in PCR tubes in a total volume of 10 μL. An aqueous solution of ODN or ORN (50 nmol) in a PCR tube was concentrated, and the ON resuspended in water (4 μL) was treated with an aqueous solution of Mg(OAc)$_2$ (2 μL, 50 mM), MES buffer (2 μL, 100 mM, pH 6.0), and an aqueous solution of diazoacetone (1 μL, 400 mM). A solution of [Rh(COD)Cl]$_2$ in THF (1 μL, 5 mM) was added and mixed thoroughly. The reaction mixture was incubated at r.t. for 0.5 h unless otherwise noted and analyzed by HPLC-MS.

Derivatization of the acetonylated products by oxime ether conjugation with benzyloxyamine was performed when partial separation of HPLC peaks was observed. The crude material from the reaction was diluted with water (20 μL), extracted with ethyl acetate (30 μL × 3), and the resulting aqueous layer was concentrated. The crude material was dissolved in water, and a small portion (5 nmol of dsDNA) was diluted with water to a final volume of 4 μL. The solution was treated with benzyloxyamine (1 μL, 50 mM in DMSO), kept at r.t. for 1 h (unless otherwise noted), and analyzed by HPLC-MS.

**Statistics and reproducibility.** All the experiments were at least duplicated to confirm reproducibility. All attempts at replication were successful.

## Data availability

The authors declare that the data supporting the findings of this study are available within the paper and its supplementary information files. Source data are provided with this paper.

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

## Acknowledgements

This work was supported by the Samsung Science and Technology Foundation (SSTF-BA1401-11) and the National Research Foundation of Korea (NRF) grants (2021R1A5A6002803 Center for New Directions in Organic Synthesis (CNOS), 2017M3A9E4078558) funded by the Korean government.

## Author contributions

C.-M.P. conceived, designed, and directed the project; Y.-H.L. performed the experiments and analyzed the data; E.Y. performed the experiments, DFT calculations, and analyzed the data; C.-M.P., Y.-H.L., and E.Y. prepared the manuscript.

## Competing interests

The authors declare no competing interests.
