## [Peer Review File · Nature Communications]

REVIEWER COMMENTS

Reviewer #1 (Remarks to the Author):

The key discovery that makes this paper exciting is the Rh(I) catalysis for carbene transfer at the O6 position of guanine. My group discovered that copper(I) carbenes can selectively form O6G adducts, but this paper provides a more efficient and high-yielding way to do these sorts of conjugations. My group struggled with long substrates and could not effectively hit bulge positions. Their method seems more reactive for O6G and guanines can be targeted in many types of unpaired scenarios (single-strand, bulges, overhangs). For only the discovery in basic reactivity this paper deserves to be accepted.

Here are some of the things I find lacking:

1. «chelation-free catalyst» - what experiments have been done to study this? To me the DFT is unconvincing as the DFT transition states are usually partially pre-defined and then optimized. How was the transition state search done? Did the authors truly sample both the chelated and unchelated possibilities? An energy comparison of these possibilities could be made, even if it requires opening a metal-binding site through olefin dissociation.
2. My (Gillingham) group used the 7-deazaguanosine as a mechanistic probe for chelation requirement. This is commercially available and a simple experiment that would test their “non-chelation” hypothesis.
3. The modification site in larger oligos was determined almost exclusively by MS/MS. While I don't doubt the correctness of the conclusion, I think the authors should include characterization of modification site in at least one of their large substrates by an additional method (restriction, polymerase extension assay). Polymerase extension would be especially informative as it can pick up low level modification more effectively than MS/MS.
4. Does the reaction work in the presence of protein?
5. Cyclic guanine dinucleotides are an important substrate class that could not be accomplished with the copper(I) method. As these molecules are important in human and bacterial biology I would recommend the authors try these as another substrate class.
6. Fig. 4 takes a full one-page spread to show that reductive amination of the ketone to a DNA binding protein (T7 RNAP) works. I would have been shocked if this part hadn't worked and I'm not sure what the authors want to accomplish here. A better demonstration of the method would be to modify the protein with the diazo compound and show that it can achieve proximity labelling of the DNA at G's close to the T7 binding site. Chip-Seq, RIP-Seq, and related methods depend on nucleic acid-protein crosslinking so testing this possibility would be more valuable for the chemical biology community.
7. I see very few RNA substrates. Given the varied biotechnological applications of modified RNA I think more attention should be paid to this substrate class.

Hence I believe the fundamental discovery in the paper is appropriate for publication in Nature communications. But I think revisions are necessary to more firmly support the hypotheses, and to convince the wider community that the tool will be useful.

Dennis Gillingham

Reviewer #2 (Remarks to the Author):

This manuscript described the chemo- and regio-selective modification of oligonucleotides by means of Rh(I)-carbene catalysis. The chemo- and regio-selective modifications of proteins and nucleic acids are important for the identification of molecules that associate in the biological events, an efficient and selective crosslinking reaction applicable in aqueous media are in great demand. The modification reaction with Rh(I)-carbene system allows the selective introduction of an acetyl group at unpaired guanosine in several secondary structures. Possible usefulness of the Rh(I)-carbene system has been demonstrated through the applications of this method to the

DNA-protein cross-linking reaction. While the optimal condition for this reaction was listed in Table 1, the logical process to optimize the suitable condition for the high yield was not described in detail. Because such processes are critical for further application of the authors' method, the process should be described in detail with appropriate discussion. For example, how the authors applied the type and contents of organic solvent (e.g. 10%, 50% THF and 20% DMSO) for a typical reaction system should be described. Judging from the reaction preference, the secondary structure of oligonucleotide is important for the regio-selectivity of this reaction. Does the secondary structure of oligonucleotide in aqueous solution containing the organic solvent predictable? Possible secondary structures of the target oligonucleotide should be analyzed with reagents that have been reported to target unpaired nucleic acid bases. The manuscript tackled an interesting application of Rh(I)-carbene system, however, the optimization process of the reaction condition and information on the possible structure of the target oligonucleotide under the particular reaction condition are very important not only for justification of the authors' notion but also for further application of the authors' method. After clarification of these issues, this manuscript could be published in Nature Communications.

Additional comments:

1. The authors noted no significant selectivity of dI against dG, but still a slight difference was observed. Is it possible to discuss the difference based on the nucleophilicity N index from DFT calculation?
2. Appropriate information on the secondary structures of these oligonucleotides in aqueous organic solvents should be analyzed by means of conventional technique. Is it necessary to use aqueous organic solvents for this reaction?
3. Why the total amount of proteins in lysate also changed from lane 1 to 4 in Fig. 4e? Judging from the experimental condition, only the T7 RNAP concentration should be changed.
4. The yield of double modification discussed in Fig. 4d was not provided and should be appeared in SI (p.33). Effect of stability of the duplex after the first modification (20b) should be discussed for the double modification.
5. The caption for Fig. 4d seems to be incorrect.

Reviewer #3 (Remarks to the Author):

This manuscript describes G-selective modification of ss-DNA and of related mismatched or otherwise unpaired sites. The science described here represents a significant jump forward in the reactivity, selectivity, and general utility of carbene reagents for selective oligo modification. The diazoacetone reagent is a remarkably simple reagent with important implications for bioorthogonal reactivity. DNA chemistry is extremely limited in breadth and scope. The reactivity demonstrated here appears truly general and utilitarian, going far beyond simple proof-of-concept experiments with very small, special sequences.

The protein cross-linking experiments (Fig 4 a-c) are an impressive demonstration of the power of these easily-obtained DNA variants. The fluorescence data (Fig 4 e) convincingly demonstrates the use in lysate. Certainly there are other ways to make non-natural oligos, but these are fairly complex constructs, and the ability to predictably i them from natural sequences is meaningful.

Publication of this manuscript seems warranted, as it could be a powerful tool to practicing scientists in the field of DNA science, and it demonstrates a remarkable fundamental advance in our understanding of controlled DNA reactivity.

I have only very limited concerns that might be addressed prior to publication.

1. Table 1 describes screening test on mixtures of nucleosides. I presume that only 1 is reactive, but this should be more clear from the table.
2. I find Fig 1 to be fairly hard to follow. In (a), the top blue arrow "contains" the monofunctionalized substrate, but it looks like it is added separately as a new additive above. Also, the two cartoon styles are confusing.
- 3.

REVIEWER COMMENTS

Reviewer #1 (Remarks to the Author):

The key discovery that makes this paper exciting is the Rh(I) catalysis for carbene transfer at the O6 position of guanine. My group discovered that copper(I) carbenes can selectively form O6G adducts, but this paper provides a more efficient and high-yielding way to do these sorts of conjugations. My group struggled with long substrates and could not effectively hit bulge positions. Their method seems more reactive for O6G and guanines can be targeted in many types of unpaired scenarios (single-strand, bulges, overhangs). For only the discovery in basic reactivity this paper deserves to be accepted.

Here are some of the things I find lacking:

1. «chelation-free catalyst» - what experiments have been done to study this? To me the DFT is unconvincing as the DFT transition states are usually partially pre-defined and then optimized. How was the transition state search done? Did the authors truly sample both the chelated and unchelated possibilities? An energy comparison of these possibilities could be made, even if it requires opening a metal-binding site through olefin dissociation.

>> We greatly appreciate the helpful suggestions and comments.

After receiving the reviewer's comment, we examined N⁷-methyl-2'-deoxyguanosine for its reactivity toward the carbene, and found that the reaction gave O⁶-alkylation in 65% yield. This result strongly supports the chelation-free model. Regarding DFT calculations, we attempted to locate the transition states of chelation models, however, were unable to get a reasonable geometry.

2. My (Gillingham) group used the 7-deazaguanosine as a mechanistic probe for chelation requirement. This is commercially available and a simple experiment that would test their "non-chelation" hypothesis.

>> Based on the comment, we attempted a reaction with 7-deazaguanosine, which failed to give the corresponding product (Supplementary Figs. 3 and 21). However, considering the result from N⁷-methyl-2'-deoxyguanosine (Fig. 2c), we reason that this failure may have to do with the intrinsic reactivity of 7-deazaguanosine, rather than the lack of N⁷ chelation.

Fig. 2 | Structure and mechanistic analysis of Rh(I)-catalyzed O⁶-G labeling. **a**, Expanded ¹H-¹³C HMBC (DMSO-_d₆) spectrum of **6**. **b**, Calculated nucleophilicity index of four canonical nucleobases (A, T, G, and C); transition state geometry of the catalyst approaching guanosine obtained by DFT calculations. **c**, Reaction performed with 7-methyl-2'-deoxyguanosine and its HPLC trace. See Supplementary Fig. 22 and p.105 for further analyses by MS fragmentation and NMR.

Supplementary Figure 3. Modification of 7-deaza-2'-deoxyguanosine. See Supplementary Fig. 21 for HPLC analysis.

Supplementary Figure 21. HPLC-MS analysis of modification of 7-deaza-dG with Rh(I)-catalysis. Elution method (solvent A: 0.1% formic acid in H₂O, solvent B: acetonitrile): 0/0-4/20-5/90-9/90-10/0 (time [min]/solvent B).

3. The modification site in larger oligos was determined almost exclusively by MS/MS. While I don't doubt the correctness of the conclusion, I think the authors should include characterization of modification site in at least one of their large substrates by an additional method (restriction, polymerase extension assay). Polymerase extension would be especially informative as it can pick up low level modification more effectively than MS/MS.

>> We included a primer extension assay with **8h** (Supplementary Fig. 6). While primer extension was not observed with purine bases (A and G, lane 1 and 4), both pyrimidine bases (T and C, lane 2, 3 and 5, 6) were incorporated to give the extended products, indicating that the act-G is recognized as either A

or G by the DNA polymerase. This is consistent with previous reports that O⁶-alkylated G elicits G to A transition.

Supplementary Figure 6. Primer extension assay with acetonilylated duplex **8h**. Lanes; L: DNA ladder, 1: with dATP at 60 min, 2: with dTTP at 40 min, 3: with dTTP at 60 min, 4: with dGTP at 60 min, 5: with dCTP at 40 min 6: with dCTP at 60 min.

4. Does the reaction work in the presence of protein?

>> Yes, we performed a reaction on oligo **7a** in the presence of lysozyme, and obtained the G-acetonilylated product **8a** in 67% yield (Supplementary Fig. 8).

a

b

c

Supplementary Figure 8. Reaction in the presence of protein. **a**, Scheme of the reaction in the presence of protein. **b**, HPLC analysis of the reaction. **c**, SDS-PAGE analysis to show the presence of protein. Lanes; 1: before reaction, 2: reaction mixture. L: protein standard ladder.

5. Cyclic guanine dinucleotides are an important substrate class that could not be accomplished with the copper(I) method. As these molecules are important in human and bacterial biology I would recommend the authors try these as another substrate class.

>> We examined cyclic guanine dinucleotide **32** for the labeling, and confirmed that it proceeds smoothly to give a mixture of mono- and bis-acetylated cyclic di-GMP **33** and **34** in 46% and 33%, respectively (Supplementary Figs. 4 and 23).

Supplementary Figure 4. Modification of c-di-GMP. See Supplementary Fig. 23 for HPLC-MS analysis.

Supplementary Figure 23. Modification of c-di-GMP with Rh(I)-catalysis

before reaction

at 1 h

33

34

Supplementary Figure 23. HPLC-MS analysis of the reaction of c-di-GMP with Rh(I)-catalysis. Asterisk(*) for found fragments. Elution method (solvent A: 5 mM TEAA, solvent B: acetonitrile); 0/0-14/20-15/90-18/90-20/0 (time [min]/solvent B).

6. Fig. 4 takes a full one-page spread to show that reductive amination of the ketone to a DNA binding protein (T7 RNAP) works. I would have been shocked if this part hadn't worked and I'm not sure what the authors want to accomplish here. A better demonstration of the method would be to modify the protein with the diazo compound and show that it can achieve proximity labelling of the DNA at G's close to the T7 binding site. Chip-Seq, RIP-Seq, and related methods depend on nucleic acid-protein crosslinking so testing this possibility would be more valuable for the chemical biology community.

>> Thank you for the suggestion. It would be an exciting exploration.

The work in Fig. 4 was performed to demonstrate the feasibility for the iterative multi-functionalization of oligonucleotides.

7. I see very few RNA substrates. Given the varied biotechnological applications of modified RNA I think more attention should be paid to this substrate class.

>> As suggested, we included two more RNA substrates in the scope (Table 2).

8i (30 min, 88%)

8j (30 min, 83%)

Hence I believe the fundamental discovery in the paper is appropriate for publication in Nature communications. But I think revisions are necessary to more firmly support the hypotheses, and to convince the wider community that the tool will be useful.

Dennis Gillingham

Reviewer #2 (Remarks to the Author):

This manuscript described the chemo- and regio-selective modification of oligonucleotides by means of Rh(I)-carbene catalysis. The chemo- and regio-selective modifications of proteins and nucleic acids are important for the identification of molecules that associate in the biological events, an efficient and selective crosslinking reaction applicable in aqueous media are in great demand. The modification reaction with Rh(I)-carbene system allows the selective introduction of an acetyl group at unpaired guanosine in several secondary structures. Possible usefulness of the Rh(I)-carbene system has been demonstrated through the applications of this method to the DNA-protein cross-linking reaction.

While the optimal condition for this reaction was listed in Table 1, the logical process to optimize the suitable condition for the high yield was not described in detail. Because such processes are critical for further application of the authors' method, the process should be described in detail with appropriate discussion. For example, how the authors applied the type and contents of organic solvent (e.g. 10%, 50% THF and 20% DMSO) for a typical reaction system should be described.

>> We greatly appreciate the valuable comments.

Regarding the solvent composition, we began the optimization on nucleoside monomers in 50% THF, which was eventually optimized to 10% THF. The use of 10% THF was for the convenience of handling the catalyst as a stock solution. We also performed a reaction in pure aqueous buffer solution, and confirmed that the small amount of THF does not affect the reaction (Supplementary Fig. 5).

For the use of 20% DMSO, it was employed once for the photocage experiment owing to the solubility of the diazo compound.

Judging from the reaction preference, the secondary structure of oligonucleotide is important for the regio-selectivity of this reaction. Does the secondary structure of oligonucleotide in aqueous solution containing the organic solvent predictable? Possible secondary structures of the target oligonucleotide should be analyzed with reagents that have been reported to target unpaired nucleic acid bases.

>> The presence of the organic co-solvent arises from the catalyst stock solution prepared in THF, which facilitates setting up the reactions. Given the reviewer's comment, we performed the labeling experiment in the absence of organic solvent, and confirmed that it showed a comparable efficiency/selectivity (Supplementary Fig. 5). Based on the observation, we reason that the presence of 10% THF would not affect the secondary structures of oligonucleotides.

Supplementary Figure 5. Comparison of G-acetylation in the absence and presence of 10% THF. Oxime ether derivatization was performed for HPLC analysis.

The manuscript tackled an interesting application of Rh(I)-carbene system, however, the optimization process of the reaction condition and information on the possible structure of the target oligonucleotide under the particular reaction condition are very important not only for justification of the authors' notion but also for further application of the authors' method. After clarification of these issues, this manuscript could be published in Nature Communications.

Additional comments:

1. The authors noted no significant selectivity of dI against dG, but still a slight difference was observed. Is it possible to discuss the difference based on the nucleophilicity N index from DFT calculation?

>> The computational results appear unlikely to explain the difference.

2. Appropriate information on the secondary structures of these oligonucleotides in aqueous organic solvents should be analyzed by means of conventional technique. Is it necessary to use aqueous organic solvents for this reaction?

>> We were not able to identify an appropriate method to confirm the integrity of the bulge sites affected by the small amount of THF. However, as mentioned earlier, we indirectly confirmed it by comparing the two reactions with and without 10% THF, where both gave comparable results (Supplementary Fig. 5). Also, the reaction works only on bulge-G's. Therefore, we hope to convince that these are sufficient evidences to support the integrity of the bulge sites.

3. Why the total amount of proteins in lysate also changed from lane 1 to 4 in Fig. 4e? Judging from the experimental condition, only the T7 RNAP concentration should be changed.

>> We kept the amount of lysate proteins the same. The reason why the intensity of the lysate bands from lanes 1 through 3 appears to be increasing would be attributed to the tailing of the T7 RNAP band owing to the increased loading. This is supported by the quantification of the bands in the gel image (standard deviation: 0.3E+06).

4. The yield of double modification discussed in Fig. 4d was not provided and should be appeared in SI (p.33).

>> Corrected as below.

The rest of the crude material from aqueous layer was further purified with the Glen-Park DNA purification cartridge. Concentrations of purified alkylated oligonucleotides were analyzed with the Thermo Scientific Nanodrop-One at 260 nm to give **22a** (70% acetylated, calculated from oxime ether conjugation). Fluorescence emission was measured by excitation at 570 nm.

Effect of stability of the duplex after the first modification (20b) should be discussed for the double modification.

>> The stability of the duplex **22** may be slightly lower than **20** owing to an additional mismatch pair, however, it does not appear to affect the reaction based on the observation that the second labeling also proceeds uneventfully.

5. The caption for Fig. 4d seems to be incorrect.

>> The figure was corrected to indicate the bulge (Fig. 5d, updated numbering).

d

Reviewer #3 (Remarks to the Author):

This manuscript describes G-selective modification of ss-DNA and of related mismatched or otherwise unpaired sites. The science described here represents a significant jump forward in the reactivity, selectivity, and general utility of carbene reagents for selective oligo modification. The diazoacetone reagent is a remarkably simple reagent with important implications for bioorthogonal reactivity. DNA chemistry is extremely limited in breadth and scope. The reactivity demonstrated here appears truly general and utilitarian, going far beyond simple proof-of-concept experiments with very small, special sequences.

The protein cross-linking experiments (Fig 4 a-c) are an impressive demonstration of the power of these easily-obtained DNA variants. The fluorescence data (Fig 4 e) convincingly demonstrates the use in lysate. Certainly there are other ways to make non-natural oligos, but these are fairly complex constructs, and the ability to predictably i them from natural sequences is meaningful.

Publication of this manuscript seems warranted, as it could be a powerful tool to practicing scientists in the field of DNA science, and it demonstrates a remarkable fundamental advance in our understanding of controlled DNA reactivity.

I have only very limited concerns that might be addressed prior to publication.

1. Table 1 describes screening test on mixtures of nucleosides. I presume that only 1 is reactive, but this should be more clear from the table.

>> Thank you very much for the helpful feedback.

To clarify, those unreactive nucleosides dC, dA, and dT were grayed out in Table 1.

Table 1 | Optimization of Rh(I)-catalyzed O⁶-G labeling

Entry	Catalyst	Time	Yield ^a	Entry	Catalyst	Time	Yield ^a
1 ^b	Rh ₂ (OAc) ₄	24 h	N/R	6 ^c	[Rh(COD)(MeCN) ₂] ₂ BF ₄	o/n	76%
2 ^b	Rh ₂ (TFA) ₄	24 h	N/R	7 ^c	[Rh(COD) ₂] ₂ OTf	o/n	48%
3 ^b	[Rh(COD)Cl] ₂	30 min	45%	8 ^c	Rh(CO) ₂ (acac)	o/n	N/R
4 ^c	[Rh(COD)Cl]₂	10 min	98%	9 ^d	[Rh(COD)Cl] ₂	3 h	N/R
5 ^c	Rh(COD)(acac)	60 min	83%				

^aBased on HPLC analysis. ^bConditions a: each nucleoside (5 mM), catalyst (5 mol%), diazoacetone (1.2 equiv.), 50% THF-H₂O, r.t. ^cConditions b: each nucleoside (5 mM), catalyst (10 mol%), diazoacetone (8 equiv.), 10% THF-H₂O, MES (20 mM), pH 6.0, r.t. ^dUsed ethyl diazoacetate instead of diazoacetone (deviation from Conditions a). N/R = no reaction. o/n = overnight.

2. I find Fig 1 to be fairly hard to follow. In (a), the top blue arrow "contains" the monofunctionalized substrate, but it looks like it is added separately as a new additive above. Also, the two cartoon styles are confusing.

>> The figure has been modified for clarification (Fig. 1a).

a

REVIEWERS' COMMENTS

Reviewer #1 (Remarks to the Author):

The authors have added experiments or logical answers to the critiques of myself and the other reviewers. I believe the manuscript is now ready for publication.

Reviewer #2 (Remarks to the Author):

The revised manuscript addressed all the concerns raised by this reviewer. The manuscript is now ready for publication.